# Alliance Decision of Supply Chain Considering Product Greenness and Recycling Competition

**Yong Liu \*, Qian-qian Shi and Qian Xu**

School of Business, Jiangnan University, Wuxi 214122, China; 6170905008@stu.jiangnan.edu.cn (Q.-q.S.); 6180903007@stu.jiangnan.edu.cn (Q.X.)

**\*** Correspondence: liuy@jiangnan.edu.cn; Tel.: +86-150-6188-9051

**Abstract:** In a closed-loop supply chain (CLSC), the right alliance can help manufacturers better manufacture green products and make more profits. Choosing the most suitable alliance partner is also critical for manufacturers. In regard to product greenness and recycling competition, this paper considers the CLSC comprised of a dominant manufacturer, a retailer, and a third-party recycler. Based on the Stackelberg game and equilibrium analysis, we discuss the optimal supply chain decision-making under four different models. Then, in order to ensure supply chain (SC) members' enthusiasm to participate in the alliance, we design a profit distribution method to distribute the total profit to SC members. The results show that manufacturer's optimal alliance decision is related to the degree of recycling competition. When less than the threshold, C alliance(the manufacturer make an alliance with the retailer and the third-party recycler at the same time) is optimal, otherwise, MR alliance(the manufacturer and the retailer make an alliance ) is more beneficial for the manufacturer.

**Keywords:** greenness; recycling competition; closed-loop supply chain; alliance decisions

---

## 1. Introduction

At present, environmental pollution issues are receiving extensive attention, many companies (such as General Electric, Body Shop, Midea) are putting a lot of effort into protecting the environment [1,2]. Actually, many developing and developed countries have implemented some policies to achieve sustainable development of resources. For example, in 1989, China promulgated the Environmental Protection Law of China, which plays an active role in improving and protecting the environment [3]. The German government enacted the Packaging Regulations in 1991, which requires product manufacturers and packaging manufacturers to undertake the classification and recycling of used packaging containers [4]. At the same time, with the increasing awareness of resource conservation and the advancement of government policies, more and more consumers are tending to buy green products for the environmental and health reasons since the start of the 20th century [5–7]. Recycling and green consumption is of great significance to resource conservation and environmental protection. It can not only help save costs and make more profits but also protect brand image and enhance brand value to some extent. Some enterprises recycle their own packing and have introduced the concept of green production after recycling, so as to achieve the purpose of saving resources and green production. For example, Kiehl's has made a contribution to environmental protection by recycling cosmetic bottles and adding green elements to make new products [8]. Starting from the brand itself, since 2011, all Kiehl's counter KCR (Kiehl's customer service representatives) have been gradually wearing special work clothes made of green textile technology. Ten plastic bottles can be made into KCR work clothes, and this not only solves the problem of empty bottles, but also reduces the clothing production costs [9]. However, there are still some problems in the recycling market, such as the fierce competition

(competition among enterprises, manufacturers, recyclers, etc.), and the high cost of recycling and producing green products [10,11]. So, how manufacturers decide the products' greenness and choose the best partners is an urgent problem. It is necessary for the closed-loop supply chain (CLSC) network to keep the balance between maximizing profit and maximizing green demand [12].

This article aims to determine the optimal alliance partner for manufacturers in the context of competitive recycling, so that their alliance can help the CLSC and manufacturers get the maximum profit. To do this, we establish four different alliances, and compare the profits of each member and CLSC under the two conditions of alliance and non-alliance. In SCs dominated by the manufacturer, based on the Stackelberg game, we can get the optimal decision. Then, we analyze the effects of recycling competition degree on the profits of the channel members and the CLSC. In this paper, as the supply chain leader, the manufacturer makes the decision first, and retailer and third-party recyclers make the decision according to the manufacturer's decision.

In particular, this paper is designed to address the following questions. First, what impact will the manufacturer's participation in the recycling activities have on the third-party recycler? Second, in the supply chain that targets the best profit, should the manufacturer form an alliance? If so, with whom? Also, how should the profit of each member in the alliance be determined so as to ensure their initiative.

This paper is organized as follows. In Section 3, this paper establishes some basic models and makes some basic decision analyses. Section 4 analyzes the optimal decision under different alliance models. Section 5 compares and analyzes the optimal decisions under different models and gives some conclusions. Section 6 designs a profit distribution method after alliance. Section 7 offers a number example and sensitivity analysis. Section 8 gives the conclusions and prospect.

## 2. Literature Review

As early as the 19th century, some scholars put forward several different concepts about optimal alliance. Iwasaki et al. believed that in many cases, the ability and resources of a single member in a system composed of multiple members are limited, the system should be divided into independent groups or alliances, and each alliance should accomplish their own sub-goal to maximize the alliance value; then, the alliance could be called the optimal alliance [13]. Referring to their definition of optimal alliance, the optimal alliance in this paper is the strategic combination among the SC members which makes the all parties' profits in the alliance increase, while the profit of the whole CLSC system is also optimal. However, this has only been studied from a qualitative perspective, such as the driving and influencing factors of alliance formation or the value of alliance [14]. By combing the existing literature, we summarize the following major factors that affect strategic alliances: resource heterogeneity, flexibility, supplier cost, and return rate [15–17]. These factors will be considered when making alliance decisions. Later, to obtain many more profits, some manufacturers are willing to establish an alliance with other members, so some scholars tried to introduce alliance into the SC [18]. They analyzed the influence of alliance on the SC with quantitative methods. They discussed the superiority of alliance and analyzed whether SC members could benefit from alliance [19–21]. By analyzing alliances of different members, they tried to find the optimal alliance type for SC members [22,23]. Their research strengthened collaboration among members. However, most SC members are self-centered because there are often various contradictions among them. So, some scholars began to study how to resolve the conflicts between retailers, suppliers, and other stakeholders when selecting the optimal alliance [24]. In order to do this, they designed a lot of coordination mechanisms to reduce or even eliminate conflicts in the alliance, such as feedback and punishment mechanisms, revenue sharing mechanism, and cooperative game theory mechanism [25–27]. Reducing supply chain conflicts through alliances in the automotive industry has been a success story. For example, Ford and GM(general motors), the big American car manufacturers, have been working with professional third-party recycling companies for years to recycle their products [28].

In recent years, with the proposal of sustainable development, consumers have begun to love the green products for their special environmental characteristics [29]. At the same time, many manufacturers save resources by recycling and remanufacturing activities. For example, Nike allying with a third-party (a nonprofit organization National Recycling Coalition) is responsible for recycling products [30]. In order to meet the market demand, they often need to make continuous green innovation to their products. However, we cannot say that the higher green level of product, the higher cost of manufacturer, so manufacturers must look for a balance between cost and greenness [12]. In a word, manufacturers face two questions: Who does the recycling? Which supply chain member should be allied with? In previous studies, there was only one recycler in the CLSC. However, there may be two or more recyclers, which results in recycling competition in the SC. In this context, some scholars have discussed the impact of recycling competition or green product on SC alliances. They considered various factors to discuss the impact of recycling competition on SC alliances, and regarded that manufacturers should recycle by themselves or outsource recycling to third parties [31,32]. For alliances under recycling competition, the existing research focuses on competition among SC members [11,33–36], competition in recycling channels [37,38], and competition between two SCs [39,40]. Also, some measures were put forward to solve the adverse effects of competition. Limiting the green level [41–43], motivating manufacturers, and establishing strategic relationships between manufacturing enterprises and logistics industry groups has promoted the development of a green CLSC [44–46]. However, this research has neglected the SC alliance with greenness and the recyclers' competition.

According to the above discussion and analysis, we found some shortcomings, as follows. (1) The existing research does not comprehensively consider the influence of competition and product greenness on alliance, which deviates from the current development trend of CLSC. (2) Most literature only studies the decision of CLSC from the perspective of recycling competition or cooperative alliance, but ignores the decision of seeking cooperation in competition. In real life, the relationship between SC members is not a simply competitive or cooperative relationship. To deal with this problem, by combining product greenness and recycling competition, we use game analysis to explore the impact of competition degree on SC decision-making, and then give the optimal alliance choice for manufacturers. In order to clarify the differences between the models proposed in this paper and the early research, the following comparison is shown in Table 1.

**Table 1.** This paper versus the literature.

| Authors | Research Perspective | | Optimal Alliance | Focus |
|---|---|---|---|---|
| | Recycling Competition | Green Level | | |
| Giovanni et al. | ✓ | | | Should the manufacturers recycle or outsource it to others? |
| Hong et al. | ✓ | | | Strategic alliances between manufacturers and retailers. |
| Huang et al. | ✓ | | | Compare competitive recycling with single recycling. |
| Wang et al. | ✓ | | | Recycling competition reduces profits. |
| Ke and Cai | ✓ | | | Incentives for manufacturers to recycle. |
| Wang et al. | ✓ | | | Manufacturers and retailers compete with remanufacturers. |
| Cao et al. | ✓ | | | Pricing and coordination. |
| Zheng et al. | ✓ | | ✓ | Manufacturer's optimal alliance. |
| Fallah et al. | ✓ | | | Simultaneous competition with Stackelberg between two CLSC. |
| Ma et al. | ✓ | | ✓ | Competition between recycling alliances. |
| Bodo et al. | | | ✓ | Drivers of alliance formation. |
| Taleizadeh et al. | | | ✓ | Factors affecting alliance decision-making. |
| Sheu et al. | | | ✓ | Compare alliance and non-alliance. |
| Nie et al. | | | ✓ | Alliances can increase the profitability. |
| Xiang et al. | | | ✓ | Cost sharing; Internet platform. |
| Fatemeh et al. | | ✓ | | Government intervention policy. |
| Reza et al. | | ✓ | | Comprehensive mathematical programming model. |
| Giovanni and Pietro | | ✓ | | Joint maximization incentives increase manufacturers' investment in green jobs and profits. |
| Ghomi-Avili et al. | ✓ | ✓ | | Fuzzy pricing model for green competition closed-loop supply chain network design. |
| Yuan et al. | | ✓ | | Improvement of SC system. |
| This paper | ✓ | ✓ | ✓ | Recycling competition; optimal alliance decision; product greenness. |

## 3. Basic Models

In reality, there is a closed-loop SC composed of manufacturers, retailers, and third-party recyclers; for example, Kiehl's and its cosmetic bottle recyclers and manufacturers. In order to make more profits and increase social welfare, some manufacturers independently recycle the used products and then introduce advanced technology to remanufacture notebooks, clothes, backpacks, and so on, which can meet the environmental protection needs [47,48]. However, green R&D (research and development) and recycling both require a lot of costs, and if the manufacturer completes these tasks independently, it will inevitably decrease profits significantly. So, manufacturers need to select an alliance partner to jointly undertake R&D and recycling tasks. Through the alliance, the stable and sustainable development of the SC can be maintained [49]. In this section, we will establish four different alliance models to analyze the decisions of manufacturers. The different alliance decisions that manufacturers can choose are shown in Figure 1.

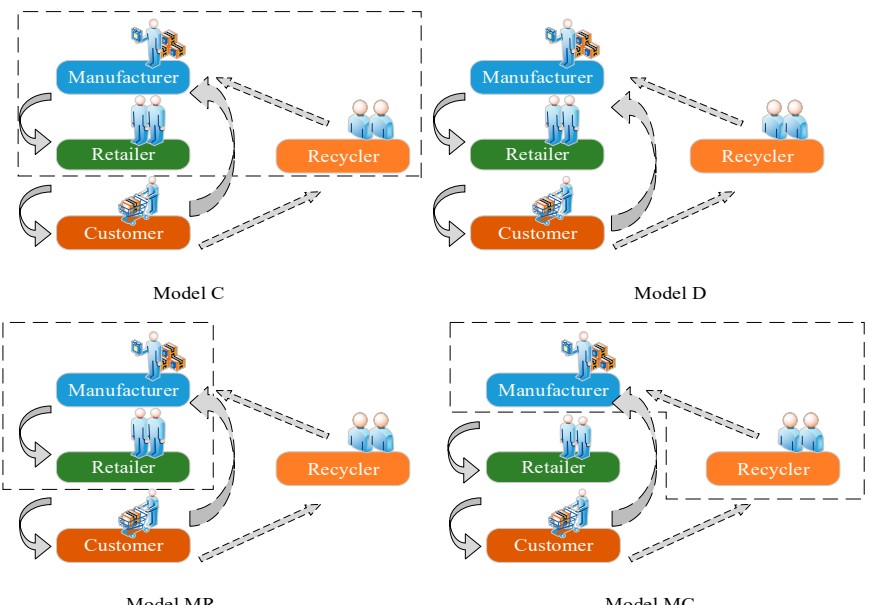

**Figure 1.** Different alliance modes. Note: C stands for centralized decision-making (the manufacture make an alliance with the retailer and the third-party recycler at the same time); D stands for decentralized decision-making (the manufacture make no alliance with any supply chain member); MR stands for the manufacturer make an alliance with the retailer; MC stands for the manufacturer make an alliance with the third-party recycler.

Figure 1 shows the four alliance models constructed in this paper. In model D, there is no alliance relationship among manufacturer, retailer, and third-party recycler. They make independent decisions, which is a decentralized decision. In model C, the manufacturer, the retailer, and the third-party recycler are in alliance at the same time, and they make decisions together, which is actually a centralized decision. In model MR(the manufacturer make an alliance with the retailer), the manufacturer and the retailer make an alliance and decide jointly, and the third-party recycler makes decisions independently. In model MC(the manufacturer make an alliance with the third-party recycler), the manufacturer allies with the third-party recycler, and the retailer makes independent decision.

In a closed-loop SC consisting of a dominant manufacturer, a retailer, and a third-party recycler, the manufacturer is responsible for remanufacturing and producing new green products. The retailer wholesales products from the manufacturer and sells them to consumers at a certain price. At the same time, the manufacturer and the third-party recycler recycle the used products from the customers, and the products recycled by the third-party recycler are delivered to the manufacturer with a transfer price.

Since this paper mainly selected the optimal alliance type for the manufacturer from the perspective of profit maximization (the profit of the manufacturer and CLSC), we assumed that the products' quality, delivery time, risk, and other factors in the different models were the same. Let $\omega, p, A, b$ represent the wholesale price, retail price, recycling price, and transfer price, respectively. Since there is no difference between new products and remanufactured products, assume that consumers are equally receptive to both of them. Assume that $c_m, c_r$ represent the manufacturer's unit cost for producing a new product and a remanufactured product, respectively, and they satisfy $c_m < c_r$. Then $\Delta = c_m - c_r$ is the unit cost saved by the manufacturer in remanufacturing. In order to ensure that manufacturer and recycler are profitable, it is necessary to meet the condition that $\Delta > b > A$.

Because consumers tend to buy green products, the higher the product greenness is, the lower the price, and the higher the sales volume of the products. In short, the product quantity ordered by the retailer is negatively correlated with sales price and positively correlated with greenness. According to the analysis, the market demand function can be written as $D = a - \alpha p + \beta g$ [50], where $a, \alpha, \beta, g$ represent the basic market demand, the consumer sensitivity coefficient of product prices, the greenness degree, and the greenness of the products, respectively, and $\alpha, \beta > 0$.

In the CLSC, there is competition between these two recycling channels. Referring to reference [33], assume that $\tau_m, \tau_c$ represent the recycling rates of manufacturer and third-party recycler, respectively, and they satisfy $I_m = \sqrt{\frac{2(I_m - \theta I_c)}{m}}$ and $I_c = \sqrt{\frac{2(I_c - \theta I_m)}{m}}$, and $0 \leq \tau_m + \tau_c \leq 1$, where $I_m, I_c, m, \theta(0 \leq \theta \leq 1)$ represent the recycling investment of the manufacturer and third-party recycler, the difficulty of recycling, and the recycling competition coefficient, respectively. The difficulty of recycling is also known as the recycling cost coefficient in some literature. The larger the value, the lower the recycling efficiency [51,52]. The recycling competition coefficient is used to measure the competition degree between recycling parties in the market. It is a kind of competition in the reverse SC [33]. The higher $\theta$, the more intense the competition between the manufacturer and third-party recycler. Therefore, the recycling cost of the manufacturer and the third-party recycler can be written as $I_m = \frac{m\tau_m{}^2 + \theta m\tau_c{}^2}{2(1-\theta^2)}$ and $I_c = \frac{m\tau_c{}^2 + \theta m\tau_m{}^2}{2(1-\theta^2)}$, respectively.

In addition, in order to improve the product greenness, manufacturers need to invest a certain amount into R&D costs. Assuming that R&D costs are quadratic with product greenness [53], then the R&D costs can be described as $I_m = \frac{1}{2}cg^2$, where $c$ represents the input coefficient of R&D costs and $c > 0$. It can be found that the manufacturer's technical input cost is a convex function of the product greenness, so excessive input cost is not economical for the manufacturer. We assumed that the decision-making period of the CLSC was a single period, and the influence of the previous cycle on current decision-making period was not considered.

In the CLSC, each member makes decisions with the goal of maximizing profits. Let $\pi_i{}^j$, $\pi^j$ represent the profit of decision maker $i$ in the model $j$ and the CLSC profit in the model $j$, where $i = T, m, r, c$, respectively, represent cooperative alliance, manufacturer, retailer, and third-party recycler, and $j = D, C, MR, MC$.

Let $\pi_m, \pi_r, \pi_c$ express the profit functions of the manufacturer, retailer, and third-party recycler, respectively, according to the above analysis, their expressions are written as follows, respectively:

$$\pi_m = (\omega - c_m)D + [(\Delta - b)\tau_c + (\Delta - A)\tau_m]D - \frac{1}{2}cg^2 - \frac{m\tau_m{}^2 + \theta m\tau_c{}^2}{2(1-\theta^2)}, \tag{1}$$

$$\pi_c = (b - A)\tau_c D - \frac{m\tau_c{}^2 + \theta m\tau_m{}^2}{2(1-\theta^2)}. \tag{2}$$

According to the above analysis, from the view of the recycling competition and product greenness, in order to solve such an alliance decision problem with a dominant manufacturer, we used a Stackelberg game (the dominant manufacturer has the priority to decide, the retailer and third-party recycler make decisions based on the manufacturer) to analyze four kinds of modes. Then, we analyzed the

optimal profit, price, and greenness degree under the different alliances, so that we could select the most profitable alliance for the manufacturer under the condition of guaranteeing product quality.

## 4. Decision Analysis Considering Product Greenness and Recycling Competition

With the increase in serious environmental problems and environmental protection awareness, manufacturers are paying more attention to the green cycle of products. The green cycle is the process of producing, selling, recycling, and re-manufacturing products. Manufacturers will consider the green preferences of consumers while producing products, and invest a certain amount to improve the product greenness. Generally speaking, the more costs invested, the greater the product greenness and the higher the price. However, multiple recyclers often compete in the process of product recycling, which may have an impact on the SC optimal decision. Therefore, the manufacturers would like to seek the best alliance to recycle the used products and develop green products. This paper analyzes the influence of different models (model D, model C, model MR, and model MC) on the optimal decision of CLSC.

### 4.1. Optimal Decision of CLSC under no Alliance (Model D)

In model D, each SC decision maker aims to maximize their own interests, and the manufacturer does not ally with any member. The decision order is that the manufacturer as the leader decides the wholesale price $\omega$, transfer price $b$, product greenness $g$, and recycling rate $\tau_m$, and then the retailer decides the retail price $p$ the third-party recycler decides the recycling rate $\tau_c$. In model D, the profit function of the manufacturer is shown as follows:

$$\pi_m{}^D = (\omega - c_m)(a - \alpha p + \beta g) + [(\Delta - b)\tau_c + (\Delta - A)\tau_m](a - \alpha p + \beta g) - \frac{1}{2}cg^2 - \frac{m\tau_m{}^2 + \theta m\tau_c{}^2}{2(1-\theta^2)}$$

$$s.t.\begin{cases} Max\pi_r{}^D = (p - \omega)(a - \alpha p - \beta g) \\ Max\pi_c{}^D = (b - A)\tau_c(a - \alpha p - \beta g) - \frac{m\tau_c{}^2 + \theta m\tau_m{}^2}{2(1-\theta^2)} \end{cases}. \tag{3}$$

**Theorem 1:** *In model D, the optimal wholesale price, transfer price, greenness, and recycling rate of the manufacturer, the optimal retail price, and recycling rate of the third-party recycler can be shown as follows:*

$$\omega^{D*} = \frac{2(\theta + 2)\alpha cc_m m - (A - \Delta)^2(\theta + 3)(\theta - 1)^2 a\alpha c - (\theta + 2)\beta^2 c_m m + 2(\theta + 2)acm}{4(\theta + 2)\alpha cm - (A - \Delta)^2(\theta + 3)(\theta - 1)^2\alpha^2 c - (\theta + 2)\beta^2 m}, \tag{4}$$

$$p^{D*} = \frac{(\theta + 2)\alpha cc_m m - (A - \Delta)^2(\theta + 3)(\theta - 1)^2 a\alpha c - (\theta + 2)\beta^2 c_m m + 3a(\theta + 2)cm}{4(\theta + 2)\alpha cm - (A - \Delta)^2(\theta + 3)(\theta - 1)^2\alpha^2 c - (\theta + 2)\beta^2 m}, \tag{5}$$

$$g^{D*} = \frac{(\theta + 2)\beta m(a - \alpha c_m)}{4\alpha cm(\theta + 2) - (A - \Delta)^2(\theta + 3)(\theta - 1)^2\alpha^2 c - (\theta + 2)\beta^2 m}, \tag{6}$$

$$b^{D*} = \frac{A(\theta + 1) + \Delta}{2 + \theta}, \tag{7}$$

$$\tau_m{}^{D*} = \frac{\alpha c(\Delta - A)(a - \alpha c_m)(\theta - 1)^2(\theta + 2)}{4\alpha cm(\theta + 2) - (A - \Delta)^2(\theta + 3)(\theta - 1)^2\alpha^2 c - (\theta + 2)\beta^2 m}, \tag{8}$$

$$\tau_c{}^{D*} = \frac{\alpha c(\Delta - A)(a - \alpha c_m)(\theta - 1)^2}{4\alpha cm(\theta + 2) - (A - \Delta)^2(\theta + 3)(\theta - 1)^2\alpha^2 c - (\theta + 2)\beta^2 m}. \tag{9}$$

**Proof:** By solving the first-order derivative of the profit functions of retailer and the third-party recycler over $p, \tau_c$, and letting $\frac{\partial \pi_r^D}{\partial p} = 0, \frac{\partial \pi_c^D}{\partial \tau_c} = 0$, we can obtain the following:

$$\begin{cases} p = \frac{\alpha\omega + \beta g + a}{2\alpha} \\ \tau_c = \frac{(\theta-1)^2(a - \alpha\omega + \beta g)(b - A)}{2m} \end{cases}. \tag{10}$$

□

Substitute (10) into the object function $\pi_m^D$, by calculating the first-order partial derivative over $\omega, g, b, \tau_m$, and setting $\frac{\partial \pi_m^D}{\partial \omega} = 0, \frac{\partial \pi_m^D}{\partial g} = 0, \frac{\partial \pi_m^D}{\partial b} = 0, \frac{\partial \pi_m^D}{\partial \tau_m} = 0$, and we can calculate the optimal decisions for manufacturer, retailer, and third-party recycler by combining it with the expression (10).

By taking (4)–(9) into object function and constraint function, then we can calculate the profits of the manufacturer, retailer, and third-party recycler and SC system as follows:

$$\pi_m^{D*} = \frac{cm(a - \alpha c_m)^2(\theta + 2)}{8\alpha cm(\theta + 2) - 2(A - \Delta)^2(\theta + 3)(\theta - 1)^2\alpha^2 c - 2(\theta + 2)\beta^2 m}, \tag{11}$$

$$\pi_r^{D*} = \frac{c^2 m^2 \alpha(a - \alpha c_m)^2(\theta + 2)^2}{\left(4\alpha cm(\theta + 2) - (A - \Delta)^2(\theta + 3)(\theta - 1)^2\alpha^2 c - (\theta + 2)\beta^2 m\right)^2}, \tag{12}$$

$$\pi_c^{D*} = \frac{\alpha^2 c^2 m(a - \alpha c_m)^2(A - \Delta)^2(\theta - 1)^2\left(1 - \theta^3 - 4\theta^2 - 4\theta\right)}{\left(4\alpha cm(\theta + 2) - (A - \Delta)^2(\theta + 3)(\theta - 1)^2\alpha^2 c - (\theta + 2)\beta^2 m\right)^2}, \tag{13}$$

$$\begin{aligned} \pi^{D*} &= \pi_m^{D*} + \pi_r^{D*} + \pi_c^{D*} \\ &= \frac{mc(a - \alpha c_m)^2\left(6\alpha cm(\theta+2)^2 - \beta^2 m(\theta+2)^2 - \alpha^2 c(A-\Delta)^2(\theta+1)\left(\theta^2 + 4\theta + 5\right)(\theta-1)^2\right)}{2\left(4\alpha cm(\theta+2) - (A-\Delta)^2(\theta+3)(\theta-1)^2\alpha^2 c - (\theta+2)\beta^2 m\right)^2}. \end{aligned} \tag{14}$$

**Theorem 2:** *If $\theta^* \leq \theta \leq 1$, then the third-party recycler will withdraw from the recycling market, and* $\theta^* = \frac{1}{6}\left(172 + 12\sqrt{177}\right)^{\frac{1}{3}} + \frac{8}{3\left(172 + 12\sqrt{177}\right)^{\frac{1}{3}}} - \frac{4}{3}.$

**Proof:** If the third-party recycler withdraws from the recycling market, there is $\pi_c^{D*} < 0$, and then $1 - \theta^3 - 4\theta^2 - 4\theta < 0$, $\theta \in [0, 1]$, so the threshold can be obtained. QED. □

According to Theorem 2, the third-party recycler and manufacturer will participate in recycling activities simultaneously only when the competition intensity is less than the threshold $\theta^*$. Otherwise, the third-party recycler will exit the recycling market. This is because the third-party's income mainly comes from manufacturer's transfer price. When competition intensity increases, the manufacturer's own recycling cost increases, which will reduce the transfer price paid to the third-party recycler. At the same time, greater competition intensity also increases the recycling cost of the third-party recycler. In summary, as competition intensity increases, the cost of third-party recycler also increases, and the profits continue to decrease until they are negative, so the third-party recycler will eventually choose to withdraw from the recycling market.

**Theorem 3:** *When $\theta^* \leq \theta \leq 1$, the optimal decisions in model D are shown as follows:*

$$\omega^{D'*} = \frac{2\alpha cc_m m - (\Delta - A)^2 a\alpha c - \beta^2 c_m m + 2acm}{4\alpha cm - (\Delta - A)^2\alpha^2 c - \beta^2 m}, \tag{15}$$

$$p^{D'*} = \frac{\alpha c c_m m - (\Delta - A)^2 a \alpha c - \beta^2 c_m m + 3acm}{4\alpha cm - (\Delta - A)^2 \alpha^2 c - \beta^2 m}, \tag{16}$$

$$g^{D'*} = \frac{\beta m(a - \alpha c_m)}{4\alpha cm - (\Delta - A)^2 \alpha^2 c - \beta^2 m}, \tag{17}$$

$$\tau_m{}^{D'*} = \frac{\alpha c(a - \alpha c_m)(\Delta - A)}{4\alpha cm - (\Delta - A)^2 \alpha^2 c - \beta^2 m}. \tag{18}$$

**Proof:** The proof is slightly similar to theorem 1.  □

So, the profits of the manufacturer, the retailer, the third-party recycler, and the systems are, respectively, given as follows:

$$\pi_m{}^{D'*} = \frac{cm(a - \alpha c_m)^2}{8\alpha cm - 2(\Delta - A)^2 \alpha^2 c - 2\beta^2 m}, \tag{19}$$

$$\pi_r{}^{D'*} = \frac{c^2 m^2 \alpha (a - \alpha c_m)^2}{\left(4\alpha cm - (\Delta - A)^2 \alpha^2 c - \beta^2 m\right)^2}, \tag{20}$$

$$\pi^{D'*} = \frac{\left(6\alpha cm - (\Delta - A)^2 \alpha^2 c - \beta^2 m\right)(a - \alpha c_m)^2 cm}{2\left(4\alpha cm - (\Delta - A)^2 \alpha^2 c - \beta^2 m\right)^2}. \tag{21}$$

**Theorem 4:** *In model D, the manufacturer's recycling volume is always greater than the third-party recycler.*

**Proof:** When the manufacturer and the third-party recycler compete for recycling in the market, there is $\frac{\tau_m{}^{D*}}{\tau_c{}^{D*}} = \theta + 2$ and $\theta \in [0, 1]$, which indicates that the recycling rate of the manufacturer is at least twice that of the third-party recycler. This is because the manufacturer dominates the SC, and the decisions of the third-party recycler are affected by the manufacturer. When the two are in a competitive position, the manufacturer will reduce the transfer price, which means the third-party recycler does not want to invest too much in recycling activities. Also, when competition between the two is fierce, the third-party is in a passive position, and then the recycling market is occupied by the manufacturer, leading to the withdrawal of the third party. This also proves the effectiveness of Theorem 2.  □

*4.2. Optimal Decisions under the Manufacturer in Alliance with the Retailer and Third-Party Recycler Simultaneously (Model C)*

Under model C, the CLSC contains the manufacturer, retailer, and third-party recycler. The CLSC members want to maximize the system profits through an alliance, so they jointly determine the retail price of the product $p$, product greenness $g$, and recycling rate $\tau_m, \tau_c$. If $0 \le \theta \le \theta^*$, then the system profit function can be expressed as follows:

$$\begin{aligned}
Max\pi^C &= \pi_m + \pi_r + \pi_c \\
&= (p - c_m)(a - \alpha p + \beta g) + (\Delta - A)(\tau_m + \tau_c)(a - \alpha p + \beta g) - \tfrac{1}{2}cg^2 \\
&\quad - \frac{m\tau_m{}^2 + \theta m\tau_c{}^2}{2(1 - \theta^2)} - \frac{m\tau_c{}^2 + \theta m\tau_m{}^2}{2(1 - \theta^2)}
\end{aligned} \tag{22}$$

**Theorem 5:** *In model C, the optimal retail price, greenness, and recycling rate of the product are shown as follows:*

$$p^{C*} = \frac{\alpha c c_m m(\theta+1) - 2a\alpha c(\Delta-A)^2(\theta-1)^2 - \beta^2 c_m m(\theta+1) + acm(\theta+1)}{2\alpha cm(\theta+1) - 2(\theta-1)^2(\Delta-A)^2\alpha^2 c - \beta^2 m(\theta+1)}, \tag{23}$$

$$g^{C*} = \frac{\beta m(a-\alpha c_m)(\theta+1)}{2\alpha cm(\theta+1) - 2\alpha^2 c(\theta-1)^2(\Delta-A)^2 - \beta^2 m(\theta+1)}, \tag{24}$$

$$\tau_m{}^{C*} = \tau_c{}^{D*} = \frac{\alpha c(\Delta-A)(a-\alpha c_m)(\theta-1)^2}{2\alpha cm(\theta+1) - 2\alpha^2 c(\theta-1)^2(\Delta-A)^2 - \beta^2 m(\theta+1)}. \tag{25}$$

**Proof:** By calculating the first-order partial derivative over $p, g, \tau_m, \tau_c$ in Equation (22), we can obtain that:

$$\begin{cases} \frac{\partial \pi^C}{\partial p} = a - \alpha p + \beta g - (p-c_m)\alpha - (\Delta-A)(\tau_m+\tau_c)\alpha \\ \frac{\partial \pi^C}{\partial g} = (p-c_m)\beta + (\Delta-A)(\tau_m+\tau_c)\beta - cg \\ \frac{\partial \pi^C}{\partial \tau_m} = (\Delta-A)(a-\alpha p+\beta g) - \frac{m\tau_m(1+\theta)}{(1-\theta)^2} \\ \frac{\partial \pi^C}{\partial \tau_c} = (\Delta-A)(a-\alpha p+\beta g) - \frac{m\tau_c(1+\theta)}{(1-\theta)^2} \end{cases}, \tag{26}$$

and then we can obtain the Hessel matrix:

$$H = \begin{bmatrix} -2\alpha & \beta & (A-\Delta)\alpha & (A-\Delta)\alpha \\ \beta & -c & (\Delta-A)\beta & (\Delta-A)\beta \\ (A-\Delta)\alpha & (\Delta-A)\beta & -\frac{m(1+\theta)}{(1-\theta)^2} & 0 \\ (A-\Delta)\alpha & (\Delta-A)\beta & 0 & -\frac{m(1+\theta)}{(1-\theta)^2} \end{bmatrix}.$$

If the function $\pi^C$ has a maximum value, then it will satisfy as follows:

$$H_1 = -2\alpha < 0, H_2 = 2\alpha c - \beta^2 > 0,$$

$$H_3 = \frac{c(\theta-1)^2(A-\Delta)^2\alpha^2 - 2cm(1+\theta)\alpha + \beta^2 m(1+\theta)}{(\theta-1)^2} < 0,$$

$$H_4 = -\frac{2(1+\theta)m\left(c(\theta-1)^2(A-\Delta)^2\alpha^2 - cm(1+\theta)\alpha + \frac{1}{2}\beta^2 m(1+\theta)\right)}{(\theta-1)^2} > 0.$$

If the function $\pi^C$ is a strict concave function, then the objective function has the optimal decision. Let Equation (26) be equal to 0 and combine it with $\pi^C$, and we can obtain the optimal retail price, greenness, and recycling rate $p^{C*}, g^{C*}, \tau_m{}^{C*}, \tau_c{}^{C*}$. QED.

By substituting $p^{C*}, g^{C*}, \tau_m{}^{C*}, \tau_c{}^{C*}$ into Equation (22), the system profit in model C can be calculated as follows:

$$\pi^{C*} = \frac{cm(\theta+1)(a-\alpha c_m)^2}{2\left(2\alpha cm(\theta+1) - 2(A-\Delta)^2(\theta-1)^2\alpha^2 c - \beta^2 m(\theta+1)\right)}. \tag{27}$$

□

### 4.3. Optimal Decisions under the Alliance of the Manufacturer and Retailer (Model MR)

In model MR, the manufacturer and retailer form an alliance to determine the retail price of the product, transfer price, product greenness, and recycling rate, and then the third-party recycler decides

the recycling rate. When $0 \leq \theta \leq \theta^*$, the objective function and the constraint conditions can be shown as follows:

$$
\begin{aligned}
Max\pi_T{}^{MR} &= (p - c_m)(a - \alpha p + \beta g) + [(\Delta - b)\tau_c + (\Delta - A)\tau_m](a - \alpha p + \beta g) - \tfrac{1}{2}cg^2 - \tfrac{m\tau_m{}^2 + \theta m\tau_c{}^2}{2(1 - \theta^2)} \\
s.t. \quad Max\pi_c{}^{MR} &= (b - A)\tau_c(a - \alpha p + \beta g) - \tfrac{m\tau_c{}^2 + \theta m\tau_m{}^2}{2(1 - \theta^2)}
\end{aligned}
. \tag{28}
$$

**Theorem 6L** *In model MR, when $0 \leq \theta < \theta^*$, the manufacturer's optimal greenness, transfer price, and recycling rate, the optimal retail price, and the optimal third-party recycling rate are shown as follows:*

$$
g^{MR*} = \frac{(\theta + 2)(a - \alpha c_m)\beta m}{(\theta + 2)(2\alpha c m - \beta^2 m) - (A - \Delta)^2 \alpha^2 c(\theta^3 + \theta^2 - 5\theta + 3)}, \tag{29}
$$

$$
b^{MR*} = \frac{A + \Delta + A\theta}{2 + \theta}, \tag{30}
$$

$$
p^{MR*} = \frac{(\theta + 2)\left(\alpha c c_m m - \beta^2 c_m m + acm\right) - (A - \Delta)^2 a\alpha c\left(\theta^3 + \theta^2 - 5\theta + 3\right)}{(\theta + 2)(2\alpha c m - \beta^2 m) - (A - \Delta)^2 \alpha^2 c(\theta^3 + \theta^2 - 5\theta + 3)}, \tag{31}
$$

$$
\tau_c{}^{MR*} = \frac{\alpha c(\Delta - A)(a - \alpha c_m)(\theta - 1)^2}{2(\theta + 2)\alpha c m - (A - \Delta)^2 \alpha^2 c(\theta + 3)(\theta - 1)^2 - (\theta + 2)\beta^2 m}, \tag{32}
$$

$$
\tau_m{}^{MR*} = \frac{\alpha c(\Delta - A)(a - \alpha c_m)\left(\theta^3 - 3\theta + 2\right)}{(\theta + 2)(2\alpha c m - \beta^2 m) - (A - \Delta)^2 \alpha^2 c(\theta^3 + \theta^2 - 5\theta + 3)}. \tag{33}
$$

**Proof:** Slightly. □

By substituting (30)–(34) into objective function and constraint function, we can obtain the optimal profits of the third-party recycler and the MR alliance, as follows:

$$
\pi_c{}^{MR*} = \frac{m(\Delta - A)^2 \alpha^2 c^2 (a - \alpha c_m)^2 \left(1 - \theta^3 - 4\theta^2 - 4\theta\right)(\theta - 1)^2}{2\left(2\alpha c m(2 + \theta) - (A - \Delta)^2 \alpha^2 c(\theta + 3)(\theta - 1)^2 - (2 + \theta)\beta^2 m\right)^2}, \tag{34}
$$

$$
\pi_T{}^{MR*} = \frac{cm(a - \alpha c_m)^2(\theta + 2)}{4\alpha c m(\theta + 2) - 2\alpha^2 c(\theta + 3)(\theta - 1)^2(\Delta - A)^2 - 2\beta^2 m(\theta + 2)}, \tag{35}
$$

and the system profit is:

$$
\pi^{MR*} = \frac{\left((2\alpha c m - \beta^2 m)(\theta + 2)^2 - (A - \Delta)^2 \alpha^2 c(\theta + 1)(\theta - 1)^2 \left(\theta^2 + 4\theta + 5\right)\right)(a - \alpha c_m)^2 cm}{2\left(2\alpha c m(\theta + 2) - (A - \Delta)^2 \alpha^2 c(\theta + 3)(\theta - 1)^2 - \beta^2 m(\theta + 2)\right)^2}. \tag{36}
$$

**Theorem 7:** *If $\theta^* \leq \theta \leq 1$, the optimal decisions in model MR will be as follows:*

$$
p^{MR'*} = \frac{\alpha c c_m m - (\Delta - A)^2 a\alpha c - \beta^2 c_m m + acm}{2\alpha c m - (\Delta - A)^2 \alpha^2 c - \beta^2 m}, \tag{37}
$$

$$
g^{MR'*} = \frac{(a - \alpha c_m)\beta m}{2\alpha c m - (\Delta - A)^2 \alpha^2 c - \beta^2 m}, \tag{38}
$$

$$
\tau_m{}^{MR'*} = \frac{(a - \alpha c_m)(\Delta - A)\alpha c}{2\alpha c m - (\Delta - A)^2 \alpha^2 c - \beta^2 m}, \tag{39}
$$

$$\pi^{MR'*} = \frac{(a - \alpha c_m)^2 cm}{4\alpha cm - 2(\Delta - A)^2 \alpha^2 c - 2\beta^2 m}.$$　(40)

**Proof:** Slightly.　□

**Theorem 8:** *The withdrawal of the third-party recycler from the recycling market has always been beneficial to the manufacturer and retailer.*

**Proof:** In model D, when third parties withdraw from the recycling market, there is $\theta^* \le \theta \le 1$:

$$\pi_m{}^{D'*} - \pi_m{}^{D*} = \frac{\alpha^2 c^2 m(a - \alpha c_m)^2 (A - \Delta)^2 (1 - \theta^3 - \theta^2 + 6\theta)}{2\Big(4\alpha cm(\theta + 2) - (\theta + 3)(\theta - 1)^2 (A - \Delta)^2 - \beta^2 m(\theta + 2)\Big)\Big(4\alpha cm - (A - \Delta)^2 - \beta^2 m\Big)}$$

When $\theta^* \le \theta \le 1$, there is $1 - \theta^3 - \theta^2 + 6\theta > 0$, that is $\pi_M^{D*'} - \pi_M^{D*} > 0$:

$$\pi_r{}^{D'*} - \pi_r{}^{D*} = \frac{a^3 c^3 m^2 (a - \alpha c_m)^2 (A - \Delta)^2 (1 - \theta^3 - \theta^2 + 6\theta)\big(8\alpha cm(\theta + 2) - (\theta^3 + \theta^2 - 4\theta + 5)(A - \Delta)^2 \alpha^2 c - 2\beta^2 m(\theta + 2)\big)}{\big(4\alpha cm(\theta + 2) - (\theta + 3)(\theta - 1)^2 (A - \Delta)^2 \alpha^2 c - \beta^2 m(\theta + 2)\big)^2 \big(4\alpha cm - (A - \Delta)^2 - \beta^2 m\big)^2} > 0$$

Similarly, in model MR, when third parties withdraw from the competitive market,

$$\pi_T{}^{MR'*} - \pi_T{}^{MR*} = \frac{c^2 \alpha^2 m(a - c_m)^2 (1 - \theta^3 - \theta^2 + 6\theta)(\Delta - A)^2}{\big(2\alpha cm(\theta + 2) - (A - \Delta)^2 \alpha^2 c(\theta + 3)(\theta - 1)^2 - \beta^2 m(\theta + 2)\big)\big(2\alpha cm - (A - \Delta)^2 \alpha^2 c - \beta^2 m\big)} > 0$$

QED.　□

### 4.4. Optimal Decision under the Alliance of the Manufacturer and Third-Party Recycler (Model MC)

In model MC, the manufacturer and third-party recycler collaborate and form an alliance. They first decide the wholesale price $\omega$, greenness $g$, recycling rate $\tau_m, \tau_c$, and then the retailer decides the retail price $p$. If $0 \le \theta \le \theta^*$, we can obtain the following expressions:

$$\begin{aligned}
&Max\pi_T{}^{MC} = (\omega - c_m)(a - \alpha p + \beta g) + (\Delta - A)(\tau_m + \tau_c)(a - \alpha p + \beta g) - \tfrac{1}{2}cg^2 - \frac{(\theta + 1)m(\tau_m{}^2 + \tau_c{}^2)}{2(1 - \theta^2)} \\
&s.t.\pi_r{}^{MC} = (p - \omega)(a - \alpha p + \beta g)
\end{aligned}. \quad (41)$$

**Theorem 9:** *In model MC, the optimal wholesale price, greenness, recycling rate, and product optimal price can be, respectively, expressed as follows:*

$$\omega^{MC*} = \frac{2\alpha cc_m m - 2(1 - \theta)(A - \Delta)^2 a\alpha c - \beta^2 c_m m + 2acm}{4\alpha cm - 2(1 - \theta)(A - \Delta)^2 \alpha^2 c - \beta^2 m},$$　(42)

$$g^{MC*} = \frac{\beta m(a - \alpha c_m)}{4\alpha cm - 2(1 - \theta)(A - \Delta)^2 \alpha^2 c - \beta^2 m},$$　(43)

$$\tau_m{}^{MC*} = \frac{\alpha c(\Delta - A)(1 - \theta)(a - \alpha c_m)}{4\alpha cm - 2(1 - \theta)(A - \Delta)^2 \alpha^2 c - \beta^2 m},$$　(44)

$$\tau_c{}^{MC*} = \frac{\alpha c(\Delta - A)(1 - \theta)(a - \alpha c_m)}{4\alpha cm - 2(1 - \theta)(A - \Delta)^2 \alpha^2 c - \beta^2 m},$$　(45)

$$p^{MC*} = \frac{\alpha cc_m m - 2(1 - \theta)(A - \Delta)^2 a\alpha c - \beta^2 c_m m + 3acm}{4\alpha cm - 2(1 - \theta)(A - \Delta)^2 \alpha^2 c - \beta^2 m}.$$　(46)

**Proof:** Slightly.  □

By bringing expressions (42)–(47) into the objective function and the constraint function, we can get the optimal profit of the retailer, the manufacturer, and the third-party recycler in model MC:

$$\pi_r^{MC*} = \frac{c^2m^2(a - \alpha c_m)^2\alpha}{4\left(2\alpha cm - (1-\theta)(A-\Delta)^2\alpha^2c - \frac{1}{2}\beta^2m\right)^2},\tag{47}$$

$$\pi_T^{MC*} = \frac{cm(a - \alpha c_m)^2}{2\left(4\alpha cm - 2(1-\theta)(A-\Delta)^2\alpha^2c - \beta^2m\right)},\tag{48}$$

$$\pi^{MC*} = \frac{(a - \alpha c_m)^2cm\left(3\alpha cm - (1-\theta)(A-\Delta)^2\alpha^2c - \frac{1}{2}\beta^2m\right)}{4\left(2\alpha cm - (1-\theta)(A-\Delta)^2\alpha^2c - \frac{1}{2}\beta^2m\right)^2}.\tag{49}$$

## 5. Comparison Among the Optimal Decisions under Different Modes

By comparing the optimal decisions in different modes, we can obtain the regular of the optimal decisions. The regular can provide a theoretical reference for the subsequent analysis. Therefore, we will compare and analyze the wholesale price, retailer price, greenness, and profit in different modes.

**Conclusion 1:** The optimal wholesale price meets the following conditions.

(1)  If $0 < \theta \le \frac{1}{2}$, then $\omega^{D*} > \omega^{D'*} > \omega^{MC*}$;
(2)  If $\frac{1}{2} < \theta \le 1$, then $\omega^{MC*} > \omega^{D'*}$.

Conclusion 1 shows that when the degree of competition is less than $\frac{1}{2}$, which indicates that the recycling competitiveness is at the dividing point, the wholesale price in model D is greater than that in model MC. In model MC, the wholesale price increases with competition intensity. When the competition intensity reaches the threshold $\theta^*$, the third-party recycler will withdraw from the recycling market in model D, and the wholesale price in model D goes down suddenly and then remains unchanged after the threshold $\theta^*$. This is because when the third-party recycler exits the recycling market, the manufacturer will recycle independently, which avoids the marginal effect of the reverse supply chain, so the wholesale price goes down. The optimal wholesale price in model MC is higher than that of no alliance in the end. Because of the lower competitive intensity, the MC alliance can increase the total recycling volume, so the manufacturer can provide the retailer with a lower wholesale price. However, when the competition intensity increases to a certain extent, the recycling cost between two parties also increases, so the wholesale price is higher. This shows that the weaker the competition intensity is, the more conducive to consumers.

**Conclusion 2:** The optimal retail price satisfies the following conditions:

(1)  If $0 < \theta \le \frac{5}{4} - \frac{\sqrt{17}}{4}$, then $p^{D*} > p^{D'*} > p^{MC*} > p^{MR*} > p^{MR'*} > p^{C*}$;
(2)  If $\frac{5}{4} - \frac{\sqrt{17}}{4} < \theta \le \frac{1}{2}$, then $p^{D'*} > p^{MC*} > p^{C*} > p^{MR'*}$;
(3)  If $\frac{1}{2} < \theta \le 1$, then $p^{MC*} > p^{D'*} > p^{C*} > p^{MR'*}$.

It can be found from Conclusion 2 that the retail prices under models MR and C are always lower than that under MC and D modes, because the manufacturer and retailer together can enhance the market information and better grasp the consumer dynamics. When the third-party recycler exits the recycling market, the retail price under model D and MR will suddenly drop, because from conclusion 1, we find that when the third-party recycler exits the recycling market, the wholesale price decreases, so the retail price reduces. In addition, when $\theta$ is small enough, the retail price in model MR is greater than that in model C; when competition $\theta$ is high, the retail price in model C is higher than that in model MR. This suggests that model C is the most beneficial to consumers when recycling

competition is small, otherwise model MR is more beneficial. This means that when the competition is low, the cooperation among the manufacturer, retailer, and third-party recycler will improve the SC decision-making efficiency. However, with an increase in recycling competition, the cooperation between the manufacturer and third-party will decrease and the contradiction will be aroused, so the decision-making efficiency will be lower than that in the MR model. So, it can be seen that when $\theta$ is smaller, the more alliance members, the higher the optimal retail price, but when $\theta$ reaches a certain value, this relationship will no longer exist.

**Conclusion 3:** The optimal greenness meets the following conditions:

(1)  If $0 < \theta \leq \frac{5}{4} - \frac{\sqrt{17}}{4}$, then $g^{C*} > g^{MR'*} > g^{MR*} > g^{MC*} > g^{D'*} > g^{D*}$;

(2)  If $\frac{5}{4} - \frac{\sqrt{17}}{4} < \theta \leq \frac{1}{2}$, then $g^{MR'*} > g^{C*} > g^{MC*} > g^{D'*}$;

(3)  If $\frac{1}{2} < \theta \leq 1$, then $g^{MR'*} > g^{C*} > g^{D'*} > g^{MC*}$.

According to Conclusion 3, when the manufacturer and third-party recycler recycle at the same time, the product greenness in model C is the largest. However, when the third party withdraws from the recycling market and the competition intensity increases to the threshold $\theta^{**} = \frac{5}{4} - \frac{\sqrt{17}}{4}$, the greenness of model MR is bigger than that of model C. This shows that before the third-party exits the recycling market, the manufacturer's optimal decision is allying with manufacturer and retailer. This is because when the competition degree is low, more alliance members can reduce the manufacturer's production cost, so that more capital can be invested in the improvement of product greenness. However, as competition becomes more and more intense, model MR will weaken the influence of the third-party recycler. Therefore, model MR has certain advantages, and the product greenness investment is also the highest.

Secondly, compare model MC and model D. When the competition degree is small, the greenness of the MC model is greater than that of the D model, but when $\theta > 1/2$, the greenness is greater in model D. Because the competition degree is relatively small, the third party participates in recycling activities, and the MC model can improve the decision-making efficiency, so the greenness degree is relatively high. When the competition degree increases to 1/2, competition reduces the decision-making efficiency of the MC model.

Combined with Conclusion 2, it can be found that the lower the retail price, the higher the greenness. This shows that the manufacturer can achieve higher green products at a lower price through alliance, which is beneficial to consumers and the environment.

**Conclusion 4:** When $\tau_m{}^{C*}$ is equal to $\tau_m{}^{MR*}$, the optimal decision is $\theta_1$, and when $\tau_m{}^{C*}$ is equal to $\tau_m^{D'*}$, the optimal decision is $\theta_2$. So, we can get following results:

(1)  If $0 < \theta \leq \theta_1$, then $\tau_m{}^{C*} > \tau_m{}^{MR*} > \tau_m{}^{MC*} > \tau_m{}^{D*}$;

(2)  If $\theta_1 < \theta \leq \theta^*$, then $\tau_m{}^{MR*} > \tau_m{}^{C*} > \tau_m{}^{MC*} > \tau_m{}^{D*}$;

(3)  If $\theta^* < \theta \leq \theta_2$, then $\tau_m^{MR'*} > \tau_m^{C*} > \tau_m^{D'*} > \tau_m^{MC*}$;

(4)  If $\theta_2 < \theta \leq \frac{\alpha c}{3\alpha c - \beta^2}$, then $\tau_m^{MR'*} > \tau_m^{D'*} > \tau_m^{C*} > \tau_m^{MC*}$;

(5)  If $\frac{\alpha c}{3\alpha c - \beta^2} < \theta \leq 1$, then $\tau_m^{MR'*} > \tau_m^{D'*} > \tau_m^{MC*} > \tau_m^{C*}$.

According to Conclusion 4, the manufacturers' recycling rate is more complex. However, it shows that if the size of the recycling competition cannot be determined, the advantage of model MR is more obvious. Compared with model C and model MR, when the competition degree is relatively small, the recycling rate of the manufacturer in model C is greater than that in model MR. This is because when the competition is low, the larger the number of alliance members, and the higher the decision-making efficiency. However, with the increase in competition degree, the recycling rate of the manufacturer in the MR model is higher than that in model C, because the higher the competition, the lower the decision-making efficiency. The change in manufacturer recycling rate in model C and model MC is consistent with the change in model C and model MR.

For model C and model D, before the critical point $\theta_2$, the recycling rate of the manufacturer in model C is greater than that in model D, and after that, the recycling rate of the manufacturer in model D is greater than that in model C. This is because model D is equivalent to decentralized decision-making, with low efficiency. However, with the increase in recycling competition, the competition between the manufacturer and third-party is intensified. More alliance members cause the opinions to be inconsistent, leading to low decision-making efficiency. The change in manufacturer recycling rate in model MC and model D is consistent with the change in model C and model D.

**Conclusion 5:** When $\tau_c^{MR*}$ is equal to $\tau_c^{MC}$, its optimal decision is $\theta_3$, and the third-party recycling rate will meet following conditions:

(1)   If $0 < \theta \leq \theta_3$, then $\tau_c^{C*} > \tau_c^{MR*} > \tau_c^{MC*} > \tau_c^{D*}$;
(2)   If $\theta_3 < \theta \leq \theta^*$, then $\tau_c^{C*} > \tau_c^{MC*} > \tau_c^{MR*} > \tau_c^{D*}$;
(3)   If $\theta^* < \theta \leq \frac{\alpha c}{3\alpha c - \beta^2}$, then $\tau_c^{C*} > \tau_c^{MC*}$;
(4)   If $\frac{\alpha c}{3\alpha c - \beta^2} < \theta \leq 1$, then $\tau_c^{MC*} > \tau_c^{C*}$.

According to Conclusion 5, if we select model C, after reaching the critical point (the third party doesn't recycle any more), the third-party recycling rate will no longer be optimal.

Comparing model MR and model MC, for the third-party recycler, when they participate in market recycling and the competition is small, since the manufacturer is in a dominant position, the alliance of the two restricts the recycling of the third-party recycler. In this case, if the manufacturer chooses to make an alliance with the retailer, the third-party recycler has enough energy to devote itself to the recycling activity, so its recycling rate in model MR is higher than that in model MC. However, when it reaches the threshold $\theta^*$, the third party needs to ally with the manufacturer, otherwise they will be forced to exit the recycling market. So, the third-party recycling rate under model MC is higher than that of model MR in the end.

For model C and mode, MC, the third party will not withdraw from the recycling market regardless of competition. However, as the number of alliance members increases, the interests of each subject become more dispersed, and the manufacturer is in the dominant position. Therefore, finally, the third-party recycling rate in model C is the lowest.

**Conclusion 6:** For alliance profits, we can find the following conclusions:

(1)   If $0 < \theta \leq \frac{5}{4} - \frac{\sqrt{17}}{4}$, then $\pi^{C*} > \pi_T^{MR*} > \pi_T^{MC*}$;
(2)   If $\frac{5}{4} - \frac{\sqrt{17}}{4} < \theta \leq 1$, then $\pi_T^{MR*} > \pi^{C*} > \pi_T^{MC*}$.

For the profit of the CLSC system, we can see that:

(1)   If $0 < \theta \leq \frac{5}{4} - \frac{\sqrt{17}}{4}$, then $\pi^{C*} > \pi^{MR*} > \pi^{MC*} > \pi^{D*}$;
(2)   If $\frac{5}{4} - \frac{\sqrt{17}}{4} < \theta \leq \frac{1}{2}$, then $\pi^{MR*} > \pi^{C*} > \pi^{MC*} > \pi^{D*}$;
(3)   If $\frac{1}{2} < \theta \leq 1$, then $\pi^{MR*} > \pi^{C*} > \pi^{D*} > \pi^{MC*}$.

According to Conclusion 6, when there are two recyclers in the market, the thresholds $\theta^{**} = \frac{5}{4} - \frac{1}{4}\sqrt{17}$, and when $\theta < \theta^{**}$, the alliance and system profits in model C are all optimal, otherwise model MR is the best alliance. When $\theta > \frac{1}{2}$, the system profit in model D is greater than that in model MC. This also shows that the third-party recycler exiting from the recycling market is beneficial to both the manufacturer and the SC system.

## 6. Revenue Distribution after Alliance

From the analysis in Section 5, we can find that when the degree of recycling competition is less than $\theta^{**}\left(\theta^{**} = \frac{5}{4} - \frac{\sqrt{17}}{4}\right)$, both the alliance profit and CLSC profit are bigger than that of model MR. When the recycling competition is greater than $\theta^{**}$, the profits of the alliance and CLSC in model

MR are bigger than that of model C. However, this only shows that model C or model MR are the optimal alliance decision from the perspective of CLSC for manufacturer. As a manufacturer, it is also necessary to ensure that the profit of each subject after the alliance is greater than that before the alliance. We assumed that the manufacturer, retailer, and third-party recycler have the same profit-sharing proportion under the four different alliance models. So, with the same sharing proportion, the more profits the system makes, the more profits each subject will receive.

Therefore, in this chapter, we need to conduct profit distribution for model C and model MR, and further determine the conditions for optimal alliance decision. Under this distribution mechanism, each SC subject shares the profit of the system in a certain proportion.

## 6.1. The Revenue Distribution in Model C

Assume that $\lambda_1, \lambda_2, \lambda_3$ represent the sharing proportion of CLSC profit distributed by the manufacturer, retailer, and third-party recycler, respectively, and $0 < \lambda_1 + \lambda_2 + \lambda_3 \leq 1$. From the view of the profit of CLSC system, when model C is optimal, in order to further ensure that model C is the optimal alliance decision for the manufacturer, it must satisfy:

$$s.t \begin{cases} \lambda_1 \pi^{C*} \geq \pi_m^{D*} \\ \lambda_2 \pi^{C*} \geq \pi_r^{D*} \\ \lambda_3 \pi^{C*} \geq \pi_c^{D*} \\ \lambda_1 + \lambda_2 + \lambda_3 = 1 \\ \lambda_1, \lambda_2, \lambda_3 > 0 \end{cases} \quad . \tag{50}$$

If we solve Equation (41), we can get,

$$A_1 \leq \lambda_1 \leq 1 - (A_2 + A_3); A_2 \leq \lambda_2 \leq 1 - (A_1 + A_3); A_3 \leq \lambda_3 \leq 1 - (A_1 + A_2).$$

Among these, $A_1 > A_2 > A_3 > 0$, and,

$$A_1 = \frac{(2+\theta)\big(2\alpha cm(\theta+1) - 2(\Delta - A)^2 \alpha^2 c(\theta-1)^2 - \beta^2 m(\theta+1)\big)}{(1+\theta)\big(4\alpha cm(\theta+2) - (\Delta - A)^2 \alpha^2 c(\theta-1)^2(\theta+3) - \beta^2 m(\theta+2)\big)},$$

$$A_2 = \frac{2\alpha cm(2+\theta)^2\big(2\alpha cm(\theta+1) - 2(\Delta - A)^2 \alpha^2 c(\theta-1)^2 - \beta^2 m(\theta+1)\big)}{(1+\theta)\big(4\alpha cm(\theta+2) - (\Delta - A)^2 \alpha^2 c(\theta-1)^2(\theta+3) - \beta^2 m(\theta+2)\big)^2},$$

$$A_3 = \frac{\alpha^2 c(\Delta - A)^2(\theta-1)^2\big(\theta^3 + 4\theta^2 + 4\theta - 1\big)\big(2\alpha cm(\theta+1) - 2(\Delta - A)^2 \alpha^2 c(\theta-1)^2 - \beta^2 m(\theta+1)\big)}{(1+\theta)\big(4\alpha cm(\theta+2) - (\Delta - A)^2 \alpha^2 c(\theta-1)^2(\theta+3) - \beta^2 m(\theta+2)\big)^2}.$$

This shows that when the supply chain profit of model C is the highest, the manufacturer can get the highest revenue according to this revenue distribution.

## 6.2. The Revenue Distribution in Model MR

Like the idea in Section 6.1, we assume that $\rho_1, \rho_2$ represent the sharing proportion of CLSC profit distributed by the manufacturer and retailer, respectively, and $0 < \rho_1 + \rho_2 \leq 1$. From the view of the profit of CLSC system, when model MR is optimal, in order to further ensure that model MR is the optimal alliance decision for the manufacturer, it must satisfy:

$$s.t \begin{cases} \rho_1 \pi^{MR*} \geq \pi_m^{D*} \\ \rho_2 \pi^{MR*} \geq \pi_r^{D*} \\ \rho_1 + \rho_2 = 1 \\ \rho_1, \rho_2 > 0 \end{cases} \quad . \tag{51}$$

If we solve Equation (51), we can get,

$$B_1 \leq \rho_1 \leq 1 - B_2; B_2 \leq \rho_2 \leq 1 - B_1.$$

Among these, $B_1 > B_2 > 0$ and:

$$B_1 = \frac{2\alpha cm(\theta + 2) - \alpha^2 c(\Delta - A)^2(\theta + 3)(\theta - 1)^2 - \beta^2 m(\theta + 2)}{4\alpha cm(\theta + 2) - \alpha^2 c(\Delta - A)^2(\theta + 3)(\theta - 1)^2 - \beta^2 m(\theta + 2)},$$

$$B_1 = \frac{4\alpha cm(\theta + 2) - 2\alpha^2 c(\Delta - A)^2(\theta + 3)(\theta - 1)^2 - 2\beta^2 m(\theta + 2)(\theta + 2)c\alpha m}{\left(4\alpha cm(\theta + 2) - \alpha^2 c(\Delta - A)^2(\theta + 3)(\theta - 1)^2 - \beta^2 m(\theta + 2)\right)^2}.$$

Also, this shows that when the supply chain profit of model MR is the highest, the manufacturer can get the highest revenue according to this revenue distribution.

In this chapter, based on the optimal alliance decision from the view of system profit maximization, we tried to determine the profit distribution of the alliance subjects and the conditions for each subject to participate actively in the alliance. Combining Sections 5 and 6, we can say that when the degree of recycling competition is less than $\theta^{**}$, model C is the best alliance; otherwise, model MR is optimal.

## 7. Numerical Illustration

According to report of Economic Co-operation and Development Organization in 2018, about 300 million tons of human-generated plastic flows into the natural environment every year. By 2050, the accumulated plastic waste in the natural environment is expected to reach 120 tons. Now, the global recycling rate of plastic waste is only 15%, and the treatment of plastic waste is particularly important. Kiehl's insists on mining raw materials based on the principle of fair trade and respect for nature. At the same time, it adheres to the concept of simple and environmentally friendly packaging and maximizes the use of recyclable materials. In 2008, Kiehl's launched a new green product that contains 100% biodegradable natural ingredients, for which all packaging materials can be recycled. The recycled cosmetic bottles can be used to make fabrics, cloth bags, and so on. Besides this, manufacturers can use the empty cosmetic bottles to produce green KCR uniforms. In order to find a balance between cost and greenness, we hope to choose the right partner for a manufacturer. So, we analyzed manufacturer's alliance decisions by specific values. It is known that the manufacturer is also involved in recycling activities while entrusting a third-party with recycling. To further prove the validity of the conclusion, we took the cost of producing a new uniform and cost of remanufacturing as $c_m = 6$ RMB/piece and $c_r = 2$ RMB/piece. Other parameters were as follows: $a = 75, \alpha = 7, m = 120, A = 2, c = 3, \beta = 3$.

According to Table 2, there is a certain threshold for competition strength, above which, the third party exits from the recycling market. When the manufacturer and the third-party recycler participate in the recycling at the same time, there is the following two results: (1) For the retailer and the third-party recycler, their profits under the alliance are greater than those without the alliance, and the total profits of the manufacturer and alliance members are greater than under no alliance. Therefore, in this case, an alliance is better than non-alliance; (2) As competition intensifies, the profits of the SC members and system are both decrease. When the competition intensifies to a certain extent, there are the following conclusions: (1) Model MC is not good for the retailer, the profits of the retailer are lower than that of non-alliance, and it is also decreasing. Moreover, the total profits in model MC are lower than in model D. Therefore, the manufacturer will not ally with the third-party recycler; (2) Regardless of competition intensity, when the manufacturer allies with the retailer, the alliance's profit and system profit are greater than that of model MC. Therefore, compared with model MC, the profits of model MR are bigger; (3) When competition intensity is large enough, the profit in model C will be smaller than that in model MR, which explains that the best alliances are not always manufacturers, retailers, and third-party alliances.

**Table 2.** Impact of competitive strength $\theta$ on manufacturer alliance selection.

| $\theta$ | D Model | | | | MR Model | | | MC Model | | | C Model |
|---|---|---|---|---|---|---|---|---|---|---|---|
| | $\pi_m$ | $\pi_r$ | $\pi_c$ | $\pi$ | $\pi_T$ | $\pi_c$ | $\pi$ | $\pi_T$ | $\pi_r$ | $\pi$ | $\pi$ |
| 0 | 24.15 | 14.99 | 0.44 | 39.58 | 63.68 | 3.04 | 66.72 | 25.05 | 16.14 | 41.19 | 70.41 |
| 0.2 | 23.19 | 13.83 | 0.007 | 37.027 | 57.44 | 0.04 | 57.48 | 24.32 | 15.21 | 39.53 | 58.82 |
| 0.4 | 23.30 | 13.96 | 0 | 37.26 | 58.13 | 0 | 58.13 | 23.63 | 14.36 | 37.99 | 53.59 |
| 0.6 | 23.30 | 13.96 | 0 | 37.26 | 58.13 | 0 | 58.13 | 22.98 | 13.58 | 36.56 | 51.01 |
| 0.8 | 23.30 | 13.96 | 0 | 37.26 | 58.13 | 0 | 58.13 | 22.36 | 12.86 | 35.22 | 49.83 |
| 1 | 23.30 | 13.96 | 0 | 37.26 | 58.13 | 0 | 58.13 | 21.78 | 12.20 | 33.98 | 49.50 |

To further verify the effectiveness of profit distribution, for model C and model MR, we took $\theta = 0.1$ and $\theta = 0.4$, respectively.

When $\theta = 0.1$, we figured out $\pi_m{}^{D*} = 23.62, \pi_r{}^{D*} = 14.35, \pi_c{}^{D*} = 0.17, \pi^{C*} = 63.35$, then we could get:

$$0.37 \leq \lambda_1 \leq 0.76, 0.23 \leq \lambda_1 \leq 0.62, , 0.01 \leq \lambda_1 \leq 0.4.$$

According to this distribution ratio, after alliance, the profits that the manufacturer, retailer, and third-party recycler can gain are at least:

$$\pi_m{}^{D'*} = 24.07 > \pi_m{}^{D*}, \pi_r{}^{D'*} = 14.57 > \pi_r{}^{D*}, \pi_c{}^{D'*} = 0.63 > \pi_c{}^{D*}.$$

When $\theta = 0.4$, we figured out $\pi_m{}^{D*} = 23.30, \pi_r{}^{D*} = 13.96, \pi^{C*} = 53.59$, then we could get:

$$0.44 \leq \rho_1 \leq 0.73, 0.27 \leq \rho_1 \leq 0.56.$$

According to this distribution ratio, after alliance, the profits that the manufacturer, retailer, and third-party recycler can gain are at least:

$$\pi_m{}^{D'*} = 23.58 > \pi_m{}^{D*}, \pi_r{}^{D'*} = 14.47 > \pi_r{}^{D*}.$$

Therefore, we found that this kind of profit distribution can help the manufacturer make more accurate alliance decisions.

In order to successfully analyze the impact of recycling competition on CLSC decisions, we made a sensitivity analysis.

Table 3 shows the numerical analysis of the optimal solutions under the four alliance models. Under different degrees of recycling competition, they show obvious regularity in increase or decrease. The numerical analysis can help us to make the sensitivity analysis more intuitively.

**Table 3.** Impact of competitive strength $\theta$ on the optimal solution under different models.

| $\theta$ | D Model | | | | | | C Model | | | |
|---|---|---|---|---|---|---|---|---|---|---|
| | $\omega$ | $p$ | $b$ | $g$ | $\tau_m$ | $\tau_c$ | $p$ | $g$ | $\tau_m$ | $\tau_c$ |
| 0 | 8.41 | 9.88 | 3.00 | 1.46 | 0.17 | 0.09 | 8.28 | 4.27 | 0.50 | 0.50 |
| 0.2 | 8.51 | 9.91 | 2.91 | 1.40 | 0.10 | 0.05 | 8.68 | 3.56 | 0.22 | 0 |
| 0.4 | 8.50 | 9.90 | 2.83 | 1.41 | 0.16 | 0 | 8.86 | 3.25 | 0.10 | 0.10 |
| 0.6 | 8.50 | 9.90 | 2.77 | 1.41 | 0.16 | 0 | 8.95 | 3.09 | 0.04 | 0.04 |
| 0.8 | 8.50 | 9.90 | 2.71 | 1.41 | 0.16 | 0 | 8.99 | 3.02 | 0.01 | 0.01 |
| 1.0 | 8.50 | 9.90 | 2.67 | 1.41 | 0.16 | 0 | 9.00 | 3.00 | 0 | 0 |
| $\theta$ | MR Model | | | | | MC Model | | | | |
| | $p$ | $b$ | $g$ | $\tau_m$ | $\tau_c$ | $\omega$ | $p$ | $g$ | $\tau_m$ | $\tau_c$ |
| 0 | 8.51 | 3.00 | 3.86 | 0.45 | 0.23 | 8.33 | 9.85 | 1.52 | 0.18 | 0.18 |
| 0.2 | 8.73 | 2.91 | 3.48 | 0.26 | 0.12 | 8.40 | 9.87 | 1.47 | 0.14 | 0.14 |
| 0.4 | 8.70 | 2.83 | 3.52 | 0.41 | 0 | 8.46 | 9.89 | 1.43 | 0.10 | 0.10 |
| 0.6 | 8.70 | 2.77 | 3.52 | 0.41 | 0 | 8.53 | 9.92 | 1.39 | 0.06 | 0.06 |
| 0.8 | 8.70 | 2.71 | 3.52 | 0.41 | 0 | 8.58 | 9.94 | 1.36 | 0.03 | 0.03 |
| 1.0 | 8.70 | 2.67 | 3.52 | 0.41 | 0 | 8.64 | 9.96 | 1.32 | 0 | 0 |

### 7.1. The Impact of the Competition Intensity on Wholesale Price and Retail Price

According to Figure 2 a and b, when the third-party recycler and manufacturer recycle the Kiehl's cosmetic bottles at the same time, the wholesale price and retail price will increase, as recycling competition intensifies. This can also be verified numerically in Table 2; when the third-party recycler withdraws from the recycling market, both the wholesale price and retail price in model D or model MR are lower than when the third-party recycler is participating in recycling competition. This is because when competition intensity increase, the cost of recycling bottles by the manufacturer increases, too. As a result, the wholesale price and retail price will both increase. When the third-party recycler is no longer participating in recycling, the manufacturer is in a dominant position in decision-making, and recycling costs will be reduced. Thus, the wholesale price and retail price are lower, and thereby we can achieve greater sales.

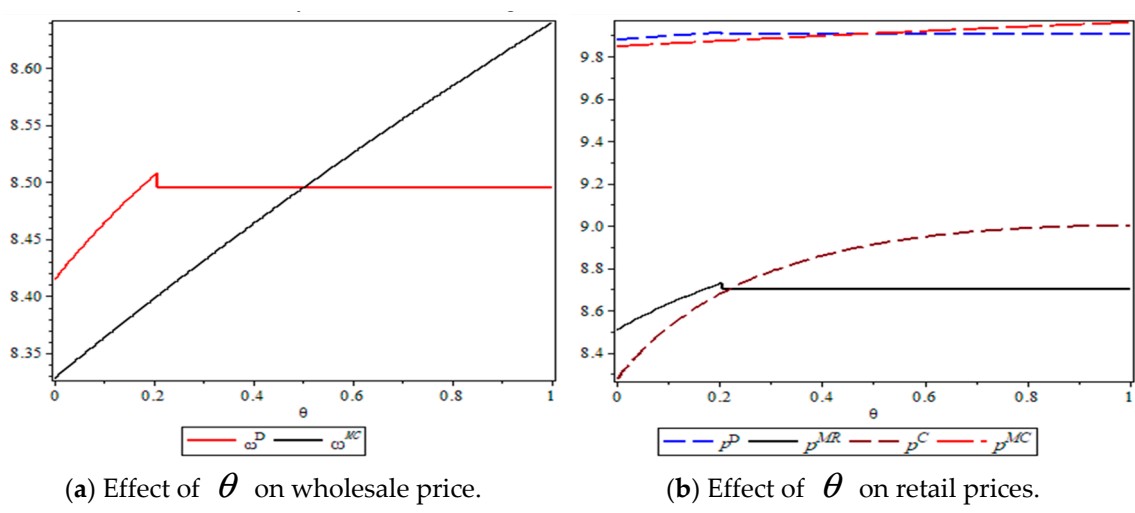

(**a**) Effect of $\theta$ on wholesale price.       (**b**) Effect of $\theta$ on retail prices.

**Figure 2.** The effect of $\theta$.

However, when competition intensity increases to a certain value, the wholesale price of model MC is higher than that of model D. It is shown in Table 3 that when $\theta = 0.4$, $\omega^{MC} < \omega^{D}$ (8.46 < 8.50). However, when $\theta = 0.6$, $\omega^{MC} > \omega^{D}$ (8.53 > 8.50). This is because the manufacturer and third-party recycler combine in order to get more revenue, but the increase in competition also bring more recycling costs, so the wholesale price and retail price of the alliances is higher than that of no alliance. Similarly, the retail price of uniforms in model C will eventually be greater than in model MR. Therefore, model C is not always better than model MR.

### 7.2. The Impact of the Competition Intensity on Transfer Price and Greenness

According to Figure 3a,b, the transfer price always decreases as the competition intensity increases, because when competition intensity is large, the manufacturer's recycling costs will increase and the transfer price paid to the third-party recycler will decrease, which is why the third-party recycler withdraws from the recycling market. At the same time, the greenness invested by the manufacturer in KCR uniforms will also decrease. Within a certain range, the product greenness in model C is greater than that in model MR. However, when the recycling competition between the manufacturer and third-party recycler is large enough, the product greenness in model MR is better than that in model C. It can be seen from Table 3 that when $\theta = 0.2$, $g^{C} > g^{MR}$ (3.56 > 3.48). However, when $\theta = 0.4$, $g^{C} < g^{MR}$ (3.25 < 3.52). This shows that the product greenness in model C is not necessarily optimal. This is consistent with the conclusions obtained in Figure 2.

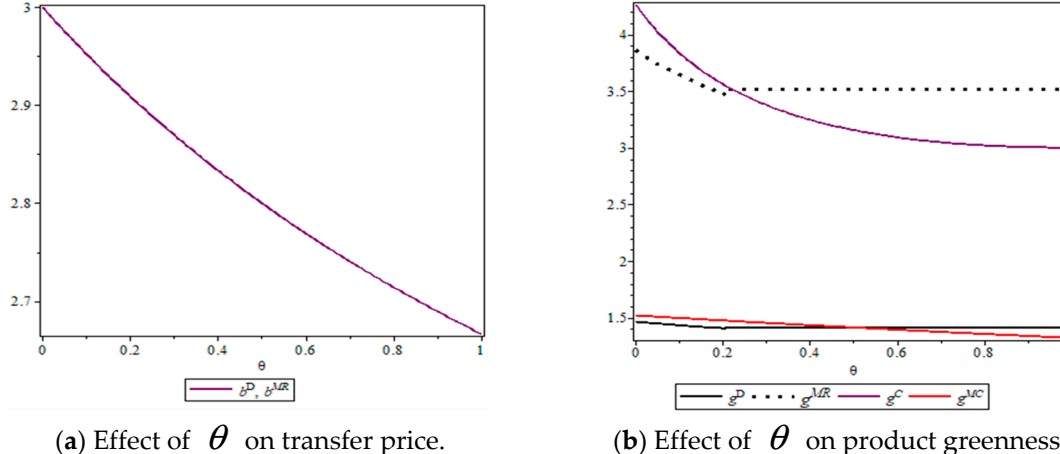

(**a**) Effect of $\theta$ on transfer price.　　　　(**b**) Effect of $\theta$ on product greenness.

**Figure 3.** The effect of $\theta$.

### 7.3. The Impact of the Competition Intensity on Recycling Rate

According to Figure 4a,b, the following conclusionscan be made. For the manufacturer: (1) When the third-party recycler competes for recycling, the manufacturer's recycling rate decreases as competition intensity increases under the four different models. This is because competition will lead to the reduction of the SC decision-making efficiency; (2) When competition intensity is low, the manufacturer's recycling rate is the highest in model C, but as competition intensity increases, when the third-party recycler has not exited from the recycling market, the manufacturer's recycling rate in model MR is higher than in model C. This phenomenon can be clearly seen in Table 3, that is, when $\theta = 0$, $\tau_m^C > \tau_m^{MR}(0.50 > 0.45)$. However, when $\theta = 0.2$, $\tau_m^C < \tau_m^{MR}(0.22 > 0.26)$. As the recycling degree increases, the difference in the manufacturer's recycling rate between model MR and model C is greater; (3) Finally, the manufacturer's recycling rate in model C is lower than that in model MC (From Table 3, when $\theta = 0.2$, $\tau_m^C > \tau_m^{MC}(0.22 > 0.14)$; however, when $\theta = 0.6$, $\tau_m^C < \tau_m^{MC}(0.04 < 0.06)$). This is because when competition intensity is low, the manufacturer is in a dominant position, and the decision-making efficiency is high when they ally with the retailer and third-party recycler at the same time. However, as competition increases, the retailer has a better grasp of the market. The alliance of the manufacturer and retailer can weaken the market influence of the third-party recycler to a greater extent. Therefore, the advantage of model MR in increasing the manufacturer's recycling rate is more obvious. However, due to the high competition intensity, despite its dominance in model C, the manufacturer needs to devote more energy to management, which greatly reduces the efficiency of the supply chain. Therefore, the recycling rate is ultimately lower than that in model MC.

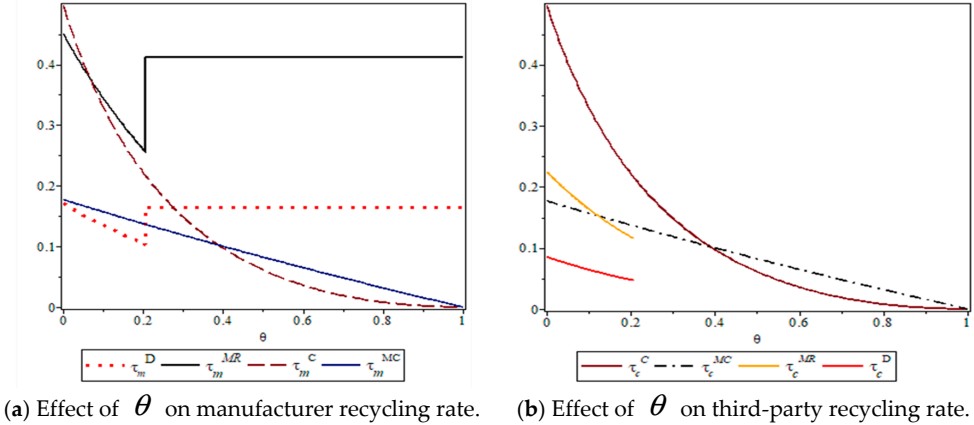

(**a**) Effect of $\theta$ on manufacturer recycling rate.　　(**b**) Effect of $\theta$ on third-party recycling rate.

**Figure 4.** The effect of $\theta$.

For the third-party recycler: (1) In the four models, the recycling rate of the third-party recycler will decrease with the increase in competition. When competition intensity is low, the third-party recycler has the highest recycling rate in model C, because increased competition will lead to higher recycling costs, so the manufacturer will reduce the transfer price paid to the third-party recycler. This results in a lack of incentive for the third-party recycler to participate in the recycling; (2) Finally, the recycling rate will be reduced. When the third-party recycler does not withdraw from the recycling activity and the competition intensity is low, the recycling rate of the third-party recycler in model MR is greater than that in model MC. However, the recycling rate of the third-party recycler in model MC is a little higher than that in model MR. This is because when competition intensity is low, manufacturers have a high degree of market information, so model MR will reduce the recycling rate of the third-party recycler. However, as competition increases, model MC can compensate for supply chain losses caused by competition, so third-party recycling rates will be higher.

### 7.4. The Impact of Competition Intensity on the Profits

According to Figure 5a–d, the profits of each SC member, the alliance parties, and the supply chain will decrease with an increase in the competition intensity, but the profit increases in non-alliance or an alliance of the manufacturer and retailer when the third-party recycler exits the market. This shows that the third-party exiting the market has always been beneficial to the manufacturer and retailer, which is consistent with Theorem 4.

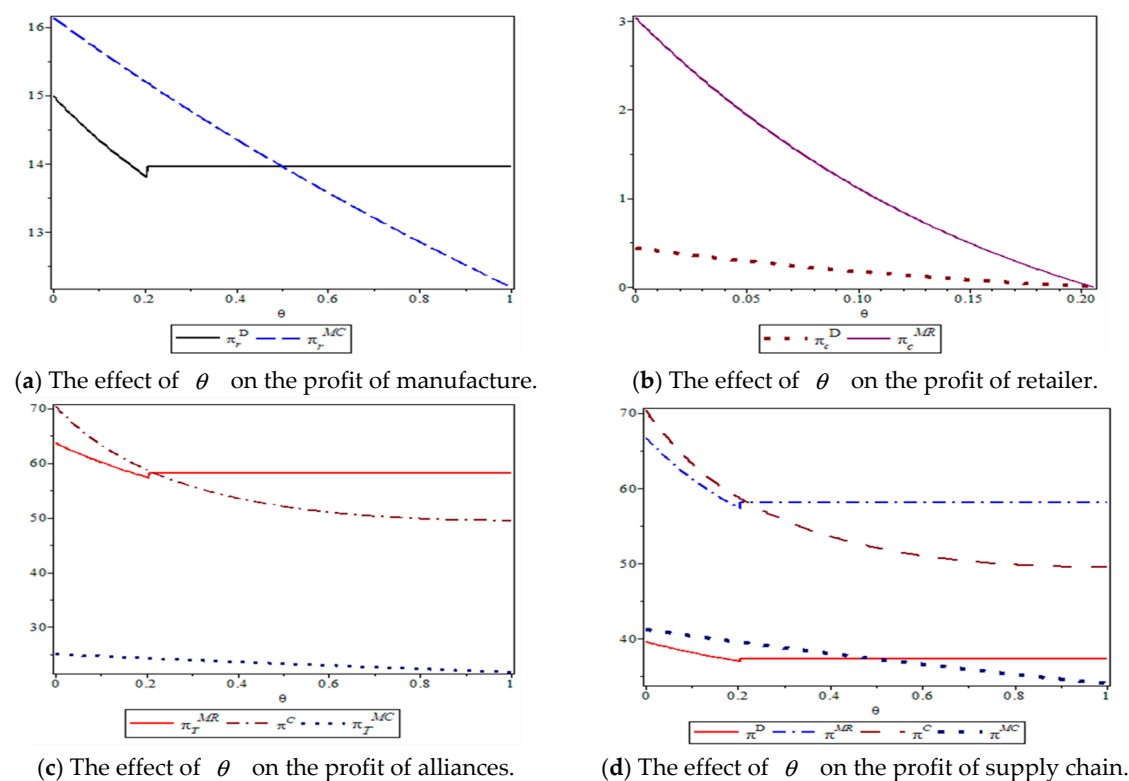

(**a**) The effect of $\theta$ on the profit of manufacture.

(**b**) The effect of $\theta$ on the profit of retailer.

(**c**) The effect of $\theta$ on the profit of alliances.

(**d**) The effect of $\theta$ on the profit of supply chain.

**Figure 5.** The effect of $\theta$.

According to the above comparative analysis, the following conclusions can be drawn: (1) The alliance of the manufacturer and the third-party recycler is uneconomical, because the MC alliance's profit is smaller than that of no alliance when competition intensity reaches a certain value, which can also be verified numerically in Table 2. From Table 2, we can see that when $\theta = 0.2$, $\pi^{MC} > \pi^D(39.53 > 37.027)$. However, when $\theta = 0.4$, $\pi^{MC} < \pi^D(37.26 < 37.99)$; (2) an alliance between the manufacturer and retailer is better than no alliance; (3) the C alliance is optimal when it is less than

the competition threshold $\theta^*$, but when greater than the threshold, the profit of model MR and the system profit will be higher than that of model C. This shows that model C is not necessarily optimal. When the manufacturer and third-party in the market participate in recycling simultaneously, the manufacturer should ally with the retailer and third-party recycler. However, when the competition intensity reaches a certain value, and the third-party withdraws from the alliance, the manufacturer should ally with the retailer. As shown in Table 2, when $\theta = 0.2 < \theta^{**}, \pi^C > \pi^{MR}(58.82 > 57.48)$. However, when $\theta = 0.4 >^{**} \theta, \pi^C < \pi^{MR}(53.59 < 58.13)$.

## 8. Conclusions and Limitations

Considering the recycling competition, we constructed four different alliance models (non-alliance, MR alliance, MC alliance, and C alliance) to find the best alliance for a manufacturer in a CLSC, and analyze the influence of recycling competition on the choice of manufacturers' alliances. Through analysis, the following conclusions were made.

(1) When the competition intensity is greater than the threshold $\theta^*$, the third-party recycler will exit the recycling market under model D and model MR, and model C is not absolutely optimal. Also, the third-party's exit is beneficial to the manufacturer, retailer, and supply chain system. Therefore, from the perspective of the third-party recycler, when the manufacturer participates in recycling and the recycling competition is relatively big, the recycler should take the initiative to seek the opportunity of allying with the manufacturer, otherwise, it should not participate in this recycling activity;

(2) When there are two recyclers in the market, the recycling rates of both the manufacturer and third-party will decrease as competition intensity increases. As competition intensity increases and the third-party does not withdraw from the recycling market, the manufacturer in the MR alliance has a higher recycling rate than that in the C alliance. This shows that the advantage of model C is only obvious when there is very small recycling competition, and when there is large recycling competition, it is more beneficial for the manufacturer to maximize profits by allying with retailers. This provides a theoretical reference for enterprises to seek the best alliance partner in the competition and make profit distribution after the alliance;

(3) When the competition intensity is less than the threshold $\theta^{**}$, an alliance of the manufacturer, the retailer, and the third-party recycler (C alliance) is the optimal decision. When higher than threshold, an alliance of the manufacturer and retailer (MR alliance) is more beneficial. For the SC managers, our conclusions provide theoretical support for the integration of the R&D of green products into the daily operation of enterprises. They can take advantage of this idea to actively seek the best alliance partners, maximize the SC performance, enhance the competitiveness of enterprises, and finally achieve the sustainable SC development.

From the above conclusions, we can find that when we take profit as the measurement standard of the alliance decision, it is unfavorable to the third-party recyclers in the case of recycling competition.

If the manufacturer does not form an alliance with a third party, the third-party recycler will eventually withdraw from the recycling market. That is to say, for the manufacturer, if they participate in recycling activities, it is uneconomic for them to entrust a special third party to carry out recycling at the same time. Through the comparison of the optimal solutions in Section 5, from the perspective of the system and the manufacturer, we found that an alliance between the manufacturer and the third-party recycler (model MC) is not the best alliance decision. The profit distribution in Section 6 ensured that the profit of each alliance member was greater than that of the non-alliance. Therefore, from the perspective of profit, the manufacturer's optimal alliance decision is related to the degree of recycling competition, but the optimal alliance is model MR or model C.

Generally, the alliance models and profit distribution constructed in this paper help manufacturers make better alliance decisions. At the same time, it also enriches the single alliance theory, from the perspective of SC, to achieve competition and cooperation. However, this paper still suffers from limitations and extends future research directions as the following:

(1) We assumed that green R&D costs are quadratic in greenness, and only consider single-phase CLSC decision-making. This may not be realistic in practice, and the cost structure may be more complex. Future studies could consider a multi-cycle CLSC and delve into the relationship between R&D costs and greenness;

(2) From the perspective of the manufacturer, this paper sets the strategic goal of the organization as the maximum profit. In fact, this is relatively narrow, so a future research direction could be developed into the optimal alliance decisions with multiple objectives (e.g., quality, cost, lead time, profit).

**Author Contributions:** Y.L. and Q.-q.S. designed and wrote this article including conceptualization, methodology, software and formal analysis, and so on. Q.X. examined the article and revised the article format.

**Funding:** This work is partially funded by the National Natural Science Foundation of China (71503103); the Humanities and Social Sciences of Education Ministry (17YJC640233); Natural Science Foundation of Jiangsu Province (BK20150157); Soft Science Foundation of Jiangsu Province (BR2018005); Jiangsu Province University Philosophy and Social Sciences for Key Research Program (2017ZDIXM034); the Fundamental Research Funds for the Central Universities (2019JDZD06); the Tender Project from Wuxi Federation of Philosophy and Social Sciences (WXSK19-A-02); Soft Science Foundation of Wuxi city(KX-19-A23); and Postgraduate Research & Practice Innovation Program of Jiangsu Provence (SJCX18_0656).

**Conflicts of Interest:** This work does not involve any conflict of interest.

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
