# Peer review of "Alliance Decision of Supply Chain Considering Product Greenness and Recycling Competition"

_sustainability, doi:10.3390/su11246900_

Round 1

Reviewer 1 Report

I suppose this an interesting issue to research the CLSC, one of the pillars of the concept of Circular Economy. I suggest authors some changes which eventually improve the study soundness, clarity and scientific quality.

improve the abstract pls, do not describe what you did, report and shortly explain conclusions from your research you declare to "use game analysis technology", I suppose this is your methodology, you eventually should explain you methodology strategy in a separate paragraph, if not your study is totally not reliable different alliance modes also need to be explained, you mention in some study points an example of cosmetics producer, I suggest to present some more to clarify modes taken into consideration your assumptions for further equations need to be much more, and more clearly, rooted in the literature thesis/findings/arrangements  and finally why your conclusions are important? for whom? what is your contribution to the theory and practice? improve your language, there are many basic but glaring errors

Author Response

We deeply appreciate the time and effort you have spent in reviewing our manuscript (sustainability-633009). Your comments are really thoughtful and helpful, we have carefully taken your kind advices and referee's detailed suggestions into consideration in revising our manuscript. Enclosure is our point-point answer to the referee’s comments. Please see the attachment. We sincerely hope this revised manuscript will be finally acceptable to be published on Sustainability. Thank you very much for all your helps and looking forward to hearing from you soon.

Reviewer 2 Report

When reviewing scientific papers for publication, I usually start with a general overview in terms of a structure, abstract, literature review, methodology, findings of the research, discussion, conclusions, as well as limitations of the study.

An interesting study that investigates alliance decision of supply chain considering product greenness and recycling competition.

The subject discussed in the paper is timely. And the structure of the paper is clear and consistent with accepted standards.

Author Response

We deeply appreciate the time and effort you have spent in reviewing our manuscript (sustainability-633009). Your comments are really thoughtful and helpful, We carefully proofread our manuscript to improve its English presentation and some details. We sincerely hope this revised manuscript will be finally acceptable to be published on Sustainability. Thank you very much for all your helps.

Reviewer 3 Report

References do not follow the MDPI style. Authors must modify them.

Title:
OK

Abstract:
Recycling is not important only for saving energy.This is a narrow vision.
Which kind of game analysis technology? Authors need to be more explicit.
Regarding optimal decisions and optimal selections, there is a lack of context and traceability.

Keywords:
Although they are included into the abstract, they are no emphasized in it.

Paper:

Introduction:
When authors write "With the rapid development of the economy and the increasing rate of resource consumption", they must be more specific giving an historical and geographical context.
Which kind of countries have implemented policies? Developed, developing, underdeveloped countries?
when do more and more consumers tend to buy organic products? There is a lack of dates, more data and references.
To protect brand image and enhance brand value to some extent is not related (is more than it) with saving the costs indicated into the abstract.
It is necessary to refer Kiehl references.
Problems in recycling market, such as the fierce competition, must be stressed and referred. The introduction section must bring out the problem to be solved with the research, but authors do not do it.
To mention the 19th century with the cite of Sambasivan and Nget,2010?. These sentences must be better expressed.
Resource heterogeneity, flexibility, supplier cost and return rate are cited, but without any kind of prioritization nor context.
Later, authors refer to obtain much more profits, which is different to save costs...authors must be homogeneus.
Coordination mechanisms in the SC alliance to reduce or even eliminate conflicts among SC members are referred, but authors do not explain the context (sector, kind of company, countries, etc.)
During the introduction, authors make a lot of statements, but very few references. They do not demonstrate nor justify nor refer many sentences. Assert-justify means you have to justify after the assertion.
This section should be divided into two groups: introduction and literature review.

Basic models:
When authors write "through the alliance, it can maintain the stable and sustainable development of the SC", for example, no references are included. This happens on numerous occasions.
Figure 1 is unreadable.
The product quantity ordered by retailer positively correlated with greenness must be better explained.
To assume that R&D costs are quadratic with product greenness (Jiang and Li, 2015) introduces limitation which must be discussed into the discussion section. Idem with the the decision-making single period of the CLSC.
The section ends and the game analysis is not explained.

3. Decision analysis considering product greenness and recycling competition:
OK. This section is the best of the paper.
But it is necessary to underline that only the cost is addressed. Other crucial paramaters like delivery, quality, or risk are not treated in this paper, so the decision is biased (at least incomplete).

4. Comparison among the optimal decisions under different modes:
The conclusions, and their implications, of this section should be better explained.
What is the translation of these formulas into the real world?
Conclusion 1 is divided into 3, 2 into 4, 3 into 4, 4 into 5, 5 into 4, and 6 into 3 ....totally 23 formulas (conclusions) that need to be better commented.

5. Numerical illustration:
Values of Cm=6 RMB/piece, Cr=2 RMB/piece and other parameters a=75, alpha=7, m=120, A=2,c=3, beta=3 should be better explained. Where do they come from?
Figure 2 is almost unreadable. Try to preset it larger and if possible avoid rasterization and pixelation.

6. Conclusions:
Although the mathematical apparatus seems solid it is necessary to emphasize that the vision is biased since other parameters, not directly related to the cost, are not treated, so the mathematical results do not help to make a decision about which is the best option.

In general, the paper needs to be improved in order to be published. Firstly, the introduction must highlight the problem, in short and long term. Secondly, a lot of assertions are made without justifying nor referring them. The mathematics is ok, but must be completed with the intervention of other key issues which are present in the multi-decision criteria: deliveries, qualities, risks, resources, requirements, etc. All these parameters would own their rules which should be harmonised with those presented in the paper, in order to make a proper decision, in a broadly sense.
Limitations are no included.
Future researches are missing.

Author Response

(The authors gave the same response as above.)

Round 2

Reviewer 1 Report

I see that authors have made many improvements, in my opinion these improvements are appropriate and making further improvements is pointless because inherent limitations of overall study design. I see this study at its current state as good enough to be published.

Author Response

We deeply appreciate you have spent the time and effort in reviewing our manuscript (sustainability-633009), and then we carefully proofread our manuscript to improve its English presentation and some details. We sincerely hope this revised manuscript will be finally acceptable to be published on Sustainability. Thank you very much for all your helps.

Reviewer 3 Report

Abstract:
Choosing the most suitable alliance partner seems to be the purpose of the search.
But how to do it?
You establish a SC among a dominant manufacturer, a retailer and a third-party recycler. Is this kind of relation representative? What is a simplification?
When authors write "this supply chain", Do you refer to the relation among a dominant manufacturer, a retailer and a third-party recycler?
Perhaps the Stackelberg game should be defined at a high level (a phrase that defines its essence and/or its main objective).
You analyze under four models? Which ones?
Is the profit the best parameter to maximize the decision?
When you write "we distribute the revenue to the members in the SC to ensure their enthusiasm to participate in the alliance", Who are you? Are you the dominant manufacturer?
This must be clarified.
The methodology of the research is absent of the abstract.
Sorry but to distribute the revenue to the members in the SC does not prove that the profit maximization of the CLSC is also the manufacturer's profit maximization.
How are optimal alliance decisions related to the degree of recycling competition? This conclusion is not supported by the previous statements.
What kind of alliances are C and MR?

Introduction:
This link ([http://blog.sina.com.cn/u/2112530335]) should be a reference.
Sorry but I am not sure that competition is the best option. There are a lot of papers researching about advantages of partnership against competition. On the other hand, reducing an organization's strategic objectives to maximum benefits generates a biased, narrow and perhaps even erroneous vision. Generating impact and securing business can achieve greater benefits in the medium to long term, as opposed to policies aimed at maximizing profit today.
However, "to determine the optimal alliance partner for manufacturers in the context of competitive recycling" is a good purpose for a research.
When authors write "In the manufacture–Stackelberg game (As the leader of the supply chain, the manufacturer makes the decision first, and retailer and third-party recycler make the decision according to the manufacturer's decision),we analyze effects of degree of recycling competition on the profits of the channel members and the CLSC", no references about this game are included.
Explanation about the game (content, purpose, proven examples of practical application, etc.) is absent.

Literature review:
Optimal alliance is not defined.
Which factors are considered in the previous literature? Which ones are included by authors in their proposal?

Basic models:
Are there not cases where the recyclers are the manufacturers themselves and/or their suppliers in the supply chain? Are they always third parties? Is this third party the best option? Is it a limitation of the research?
More examples like Kiehl should be included.
When authors write "if the manufacturer completes these tasks independently, it will inevitably decrease profits significantly", it means that a third party is always cheaper. But the recycling may be a competitive advantage itself, and perhaps in those cases it is more interesting that this knowledge lies in the manufacturer. It depends on the "garbage" to be recycled.
Four models are presented: C, D, MR and MC. Are there more models? Why are these 4 chosen? What criteria are included for their selection? Is this contrasted? For example, the approach to join retailer and recycler is not considered. And the breakdown of the manufacturer into its SC is not included.
Introduction refers to SC, but in the Figure 1, the SC is not considered.
w, p, A,b are prices and Cm and Cr are costs.
Authors assume the linear correlations of "Zhu, W.G.; He, Y.J. Green product design in supply chains under competition. European Journal of Operational Research. 2017, 258, 165-180." Authors assume the quatratic correlations of "Jiang, S.Y.; Li, S.C. Green supply chain game model and revenue sharing contract considering product greenness. Chinese Journal of Management Science.2015, 23,169-176.
Concepts like "difficulty of recycling" and "intensity coefficient of recovery competition" are mathematically defined but they are not explained nor justified.
The Stackelberg game is used as a tool for the analysis of the four models but it is not justified.
All the equations presented are based on establishing ratios and relationships between prices, costs, sales and profits. All sustainability is reduced to a matter of (direct) economy.
It should be better justified, the discourse should have been supported by a greater number of references, and should have been led unfailingly towards a three-part solution, in which economic criteria and not other strategic objectives are considered.

Decision analysis considering product greenness and recycling competition:
Teorems are interesting only if the context supports them. If those considerations are being real, if companies are acting according to these principles, if limitations of this approach is well defined.

Comparison among the optimal decisions under different modes:
Concepts like "degree of competition is less than 1/2" should be explained not only in a mathematically way.
The conclusions obtained must own a real meaning.

Revenue distribution after alliance:
Model C is the best alliance and model MR is optimal, once contextual conditions (relation among SC, strategical objectives, etc.) are defined.
This is a limitation that must be considered at the end of the paper.

Numerical illustration:
This example should be better introduced. In fact, previous sections should indicate that "an numerical example will be exposed in section 7" or similar.
Figures 3-5 are not numerically explained.

Conclusion:
There is no discussion of results. This is a problem because it makes it difficult to understand the applicability of the results. What happens with other alliances? What happens when you pursue other strategic objectives beyond the purely economic ones?
Is there only one conclusion? The title of the section should be in plural.
Limitations and future research are not considered.

Author Response

We deeply appreciate you have spent the time and effort in reviewing our manuscript (sustainability-633009). Your comments are really thoughtful and helpful, we have carefully taken your kind advices and referee's detailed suggestions into consideration in revising our manuscript. Enclosure is our point-point answer to your comments, please see the attachment. Thank you very much for all your helps and looking forward to hearing from you soon.

Round 3

Reviewer 3 Report

I thank authors the effort made.

I really think the paper is now better and the research described is more robust.

Congrats.